# A Devil of a Transmissible Cancer

**DOI:** 10.3390/tropicalmed5020050

**Published:** 2020-04-01

**Authors:** Gregory M. Woods, A. Bruce Lyons, Silvana S. Bettiol

**Affiliations:** 1Menzies Institute for Medical Research, College of Health and Medicine, University of Tasmania, Hobart, TAS 7000, Australia; 2Tasmanian School of Medicine, College of Health and Medicine, University of Tasmania, Hobart, TAS 7000, Australia; bruce.lyons@utas.edu.au (A.B.L.); s.bettiol@utas.edu.au (S.S.B.)

**Keywords:** devil facial tumor disease, parasite, transmissible cancer, MHC, immune escape

## Abstract

Devil facial tumor disease (DFTD) encompasses two independent transmissible cancers that have killed the majority of Tasmanian devils. The cancer cells are derived from Schwann cells and are spread between devils during biting, a common behavior during the mating season. The Centers for Disease Control and Prevention (CDC) defines a parasite as “An organism that lives on or in a host organism and gets its food from, or at, the expense of its host.” Most cancers, including DFTD, live within a host organism and derive resources from its host, and consequently have parasitic-like features. Devil facial tumor disease is a transmissible cancer and, therefore, DFTD shares one additional feature common to most parasites. Through direct contact between devils, DFTD has spread throughout the devil population. However, unlike many parasites, the DFTD cancer cells have a simple lifecycle and do not have either independent, vector-borne, or quiescent phases. To facilitate a description of devil facial tumor disease, this review uses life cycles of parasites as an analogy.

For more than 50 years, Professor Goldsmid, in his memorable lectures to undergraduate students, would eloquently elaborate on the features of parasites. Some features could be attributed to cancer cells. Parasites are present in almost all species and parasitism is a hugely successful life form [1]. Cancer can be found in a variety of species [2]. Analogous to a parasite, cancer cells access various critical growth and survival resources and acquire essential nutrients from the physiology of the host [3,4]. Parasites exist within a species for most of their lifecycle and exploit the host to obtain nutrients [5]. However, in contrast to cancer, a parasite will establish a symbiotic relationship with its host by “sharing” metabolites, but not necessarily exhausting the host of its supply [5]. This review describes the life history of devil facial tumor disease (DFTD) using the analogy that DFTD can display some parasitic-like features.

Devil facial tumor disease (DFTD) was serendipitously first observed by a photographer. In 1996, Christo Baars, an amateur wildlife photographer, traveled to Tasmania, the island state of Australia, which is one of the world’s most southerly islands. At 40° south, Tasmania is not a tropical island, but one where parasitic diseases have been diagnosed [6]. Christo Baars had difficulty in locating devils, but the photographs he took showed devils with grossly deformed lumps around their faces. It was possible that lesions were caused by a virus or a parasite. It was not until 2006 that results of scientific research documenting DFTD appeared in the literature [7,8,9,10]. The DFTD cancer cells displayed substantial chromosomal abnormalities. Remarkably, all DFTD cancer cells shared the same multiple chromosome abnormalities. As the karyotypes were too consistent to be considered coincidental, it led to the now accepted hypothesis that DFTD is an infectious cancer, rather than caused by a virus or parasite [10].

In 2014, a second, and independent, transmissible cancer was detected in south-east Tasmania [11]. Consequently, DFTD comprises two independent transmissible cancers, DFT1 (first identified in 1996) and DFT2 (first identified in 2014). DFT2 is currently restricted to a small pocket in the southeast of the island, whereas DFT1 has been transmitted across most of mainland Tasmania. Throughout this review, DFTD will refer to both DFT1 and DFT2; DFT1 will refer to the devil facial tumor disease first observed in 2006 and DFT2 will refer to the devil facial tumor disease first observed in 2014.

The infectious nature of the DFTD cancer cells is consistent with an infectious parasitic disease, but the parasite is a cancer cell, not an infectious microorganism. DFTD cancer cells are transmitted to another Tasmanian devil, re-establishing cancer, and then the transmission process is repeated. The DFTD cancer cells could be considered analogous to a parasite as they share other features. The suggestion that DFTD is a parasitic disease was proposed at a free public event in 2012 hosted by the Australian Society for Parasitology and Inspiring Australia [12]). Ujvari and colleagues suggested that “DFTD should be considered an evolving parasite that, like parasites, can alter life-history traits” [13,14]. For DFTD to be considered a parasite, specific criteria must be satisfied. The Centers for Disease Control and Prevention (CDC) defined a parasite as “An organism that lives on or in a host organism and gets its food from or at the expense of its host.” [15]. From this simple definition, DFTD and some parasites share similar features.

DFTD cancer cells (DFT1 and DFT2) establish a cancer mass in the oral and/or facial regions, causing gross facial deformities (Figure 1A,B). Leishmania can cause a parasitic infection that produces a facial disfigurement. However, the facial disfigurement caused by Leishmania is usually a result of inflammation, whereas the facial disfigurement associated with DFTD is a combination of the cancer mass and inflammatory response, including ulceration (as shown in Figure 1).

Concerning transmission, DFTD cancer cells benefit from the aggressive behavior of Tasmanian devils, a common occurrence during the mating season. Many parasites can be directly transmitted from diseased hosts to healthy recipients, also during the mating season. However, it is often the act of coitus, rather than biting, that allows direct parasite transmission [16]. With Tasmanian devils, however, it is biting that facilitates cancer cell transmission. The biting behavior of devils can be an aggressive ritual, with many of the bites occurring on the face [17]. These bites can cause penetrating wounds, ideal for inoculation. Ironically, the most reproductively “fit” devils are those more likely to become infected [18]. In contrast, diseases caused by parasites such as *Cryptosporidium* are linked to the immune status of the host. Although exposure may not discriminate, cryptosporidiosis primarily affects immunocompromised hosts [19]. The biting injuries that devils receive suggest that the dominant (fit) individuals are mostly responsible for transmission. Dominant devils have a higher incidence of DFT1 than submissive devils. Consequently, as the initial tumors are more likely to be inside the oral cavity, it is feasible that the dominant individuals are biting into the tumors of diseased devils [20]. The dominant and now diseased devils could then transmit DFTD when they bite a submissive devil on the face, thereby establishing a continuous mode of transmission. A mode of transmission is an essential prerequisite for any parasite. Devil biting occurs among males, among females, and between males and females. It is mostly the adult devils, but occasionally the sub-adult devils, that are aggressive. Via biting, DFTD is rapidly spread throughout the mature male and female devil population. Similarly, rabies is transmitted through biting [21]. In contrast, parasite transmission through biting usually requires an intermediate vector. Examples in humans include the anopheles mosquitoes transmitting malaria [22] and sandflies transmitting leishmaniasis [23].

Within ten years following the first observation, DFT1 spread from one devil to 51% of the Tasmanian devil population [7]. Within 20 years, the devil population had declined by approximately 80% [24]. Some devil populations lost 95% of their individuals [24,25]. During this time, it appeared that all devils with DFTD would die within 12 months [26]. This is contrary to many parasitic diseases where the parasite does not always kill its host and could be considered as “cohabitants” [27]. However, devils with DFT1 survive long enough to allow the transmission of some cancer cells to a healthy host, allowing perpetuation of the DFT1 cancer cell lineage. With limited evidence of resistance to DFT1 and the disease sweeping through the devil population, early estimates predicted that extinction could occur within 25–35 years since the first recorded case [25].

Extinction is not an ideal situation for any parasite as loss of the parasite’s sole host would correlate to extinction of the parasite. Evidence has been gradually accumulating that some Tasmanian devils can show an immune response to DFTD. Signs of recovery from DFT1, with an associated immune response, were reported in four devils [28], indicating that although recovery from DFT1 is a rare event, it does occur. By comparing DNA from Tasmanian devils collected before DFT1 arrival with DNA collected from devils collected after DFT1 arrival, Epstein and colleagues proposed that devils are rapidly evolving in response to DFT1 [29]. Evidence for recovery from DFT1 combined with the proposal of rapid evolution of devils to DFT1 provides hope for the long-term persistence of Tasmanian devils. This latter point has been proposed following mathematical modeling of devil populations, up to ten years following DFT1 emergence. The simulation modeling predicted a 21% possibility of devil extinction within 100 years following DFT1. There was a 22% chance of devils living with DFT1 and a 57% chance that DFT1 would disappear [30]. However, this study was only based on DFT1 and did not consider the impact of the second cancer, DFT2. The key message is that current evidence does not support extinction of the Tasmanian devil, thus providing “security” for the long-term presence of DFTD (DFT1 and/or DFT2). Consequently, the parasitic-existence of DFTD and co-evolution with Tasmanian devils will be maintained for the next 100 years. Long-term existence is a critical pre-requisite for any parasite. It is possible that, when a parasite first infected a species, death occurred in almost 100% of the hosts. Gradually, evolution of the host and parasite occurred, allowing the host and parasite to co-exist in equilibrium. It is possible that we are witnessing the early stages of co-evolution of Tasmanian devils and DFTD.

For DFTD to be analogous to a parasite, it would have to be the DFTD cancer cells that are transmitted, rather than a virus (e.g., papillomavirus), that independently induced cancer. The transmitted DFTD cancer cells would establish a cancer in the new host. If this were the case, the cancer cells in the new host devil would be identical to the DFTD cancer cells in the original host devil. As discussed above, the consistency of the chromosomal changes meant that it would be too much of a coincidence for the DFT1 cancer cells to share similar complex chromosomal rearrangements if DFT1 had arisen independently in every diseased devil [31,32,33]. Other studies that support that the DFT1 cancer cells are the etiological agents and thus transmitted between devils include microsatellite and major histocompatibility complex class I (MHC-I) genotyping of host and DFTD cancer cells [34,35]. The potential discovery of a second transmissible cancer, DFT2, provoked an immediate analysis. A thorough examination confirmed the cancer cells to be the aetiological agent and, therefore, DFT2 was a second transmissible cancer [11,36]. The conclusion from all the molecular and genetic studies of DFT1 and DFT2 is that host tissue DNA and cancer cell DNA are different. The DFTD cancers could not have arisen from any host tissue. Such a situation with the DFTD cancer cells is analogous to the transmission of parasitic diseases, as a parasite is not derived from host tissue.

Once DFTD cancer cells or parasites are transmitted to a new host, the cells are confronted with an almost impenetrable barrier. The host’s immune response is a major reason that transmissible cancers are extremely rare. In contrast, parasite transmission is far from rare, as parasites have developed strategies to subvert the host’s immune system (Table 1). For DFTD cancer cells to avoid recognition by the host’s immune system, some of the strategies outlined in Table 1 need to be adopted. Although it is possible for the devil’s immune system to produce an immune response to DFTD cells [37], wild devils with DFTD rarely produce an immune response against the transmitted cancer cells [38]. This is despite Tasmanian devils having a competent immune system [39,40,41]. Consequently, the DFTD cancer cells must have developed effective immune escape mechanisms.

A perusal of Table 1 provides potential mechanisms employed by parasites that DFTD cancer cells could also use to avoid the host’s immune response. Investigations into the immune system of the Tasmanian devil have not identified immune deficiencies that could explain how DFTD cells can be transmitted without inducing an immune response. Within the limitations of reagent availability, a consequence of working with a unique species, assessment of lymphoid architecture, and cellular and humoral immune responses revealed a competent immune system [39,40,41,43]. The toll-like receptors of devil innate immune cells are functional, allowing recognition and reaction to a range of pathogens, providing evidence for a competent innate immune system [44]. Immunized devils can produce an immune response to DFTD cells [37,45], activated lymphocytes can kill DFTD cells, and skin grafts are rejected [46]. Therefore, deficiencies in allorecognition mechanisms and anti-tumor immunity do not explain why the transmitted DFTD cancer cells establish in the recipient devil. Immunocompromised hosts are susceptible to some parasitic infections, such as *Cryptosporidium* [47,48]. Furthermore, and unlike helminth parasites [49], it is unlikely that DFTD cancer cells suppress the host’s immune system in order to establish. This is because Tasmanian devils with DFT1 appear to have an immune system comparable to healthy devils [39]. DFTD cancer cells do not behave like the helminth parasites by suppressing the immune system, or like *Cryptosporidium* by affecting immunosuppressed hosts. The DFTD cancer cells, similar to parasites, must have developed mechanisms to escape the host’s immune response. However, DFTD cancer cells have developed different immune escape mechanisms to parasites.

The vast majority of devils with DFTD do not show evidence for an immune response to the DFTD cancer cells. The most likely immune escape mechanism would be the capacity of the DFTD cancer cells to avoid immune recognition. The DFTD (DFT1 and DFT2) cancer cells are eukaryotic, derived from devil tissue, and Schwann cell in origin [50,51]. Therefore, they do not need to employ sophisticated immune avoidance strategies. DFTD cells only need to avoid allogeneic immune recognition. Eukaryotic cells express molecules of the major histocompatibility complex (MHC). There are two classes; MHC class I (MHC-I) and MHC class II (MHC-II). The relevance of MHC-I and MHC-II is that they represent the final step of antigen processing and present antigen peptide to T cells.

Avoidance of antigen processing and antigen presentation to T cells provides an effective escape mechanism. Some parasitic protozoa can cleverly manipulate antigen presentation to avoid inducing an immune response [52]. To circumvent antigen processing and presentation and avoid immune recognition, DFT1 cancer cells employ a simple strategy; DFT1 cancer cells do not express MHC molecules [53]. Epigenetic downregulation of critical MHC processing genes prevents the MHC molecules from being expressed on the surface of the DFT1 cancer cells. As no DFT1 antigens are presented to T cells, the DFT1 cancer cells are effectively “invisible” to the host’s immune system. Although the absence of MHC expression is a simple strategy, a similarity to some parasites is that the mechanisms accounting for MHC-I downregulation are complex. Histone deacetylase appears to epigenetically silence genes such as β_2_m, TAP1, and TAP2, thereby preventing MHC-I expression on the DFTD cancer cells. As the antigen processing genes are present, but downregulated, their expression can be restored following exposure to trichostatin A (TSA) or interferon-γ (IFN-γ) [53]. Further analysis of the antigen processing pathway revealed that the ERBB–STAT3 axis was activated in DFT1 cancer cells [54]. The activated genes in the ERBB–STAT3 axis had two effects: the promotion of cell growth and the downregulation of MHC-I. The complexity of MHC-I downregulation was exemplified following a genome-wide clustered regularly interspaced short palindromic repeats (CRISPR)-CRISPR-associated protein (Cas)9 (CRISPR/Cas9) screen of DFT1 cancer cells. The epigenetic silencing of the MHC-I processing pathway in DFT1 was also related to the polycomb repressive complex-2 (PRC2) [55].

Upregulation of MHC-I expression on the DFTD cancer cells has provided a useful strategy towards a vaccine or immunotherapy [56,57]. Similar to many parasitic diseases, such as malaria [58], producing an effective vaccine has proved elusive. The first phase 3 clinical trial of a malaria vaccine was partially effective as it prevented approximately 40% of malaria cases in children during the four-year follow-up [59]. A prototype vaccine using killed DFTD cancer cells has only been partially effective (Pye, R; personal communication). The pathway to an effective vaccine to protect against DFTD may require strategies that have been attempted with parasites such as isolating T cell epitopes for malaria [60] or a combination with drugs that interfere with the ERBB–STAT3 axis [54].

Theoretically, the absence of MHC-I expression on the membrane of DFTD cancer cells would make the cells targets for natural killer (NK) cells. Despite genetic and immunohistochemical evidence for the presence of NK cells [61,62], there is no verification for spontaneous NK cell responses to DFTD cancer cells [61]. Mitogen-activated peripheral blood lymphocytes can kill DFTD cancer cells. Therefore, NK cell activation appears to be prevented by the DFT1 cancer cells. A strategy used by *Plasmodium falciparum* is that, following infection of red blood cells, the *Plasmodium falciparum* produces inhibitory receptors that are expressed on the infected red blood cells [63]. These inhibitory receptors, collectively known as a repetitive interspersed family (RIFIN), bind to B cells and NK cells, and prevent activation, thereby preventing NK cell activation and protection of the parasite. It is unknown why DFTD cancer cells fail to activate NK cells, but they may express inhibitory molecules, and thus utilize a similar strategy to *Plasmodium falciparum*. The cancer cells of the second transmissible cancer, DFT2, express non-classical MHC molecules [64], which are known to inhibit T cell and NK cell function.

The above provides support for classifying the DFTD cancer cells as parasites, but one key element is missing. Parasites usually have a complex life cycle that can involve more than one host. The life cycle of Leishmania involves humans and sand flies, and the formation of amastigotes and promastigotes. *Toxoplasma gondii* infects humans and cats, and the life cycle involves tachyzoites, bradyzoites, sporozoites, trophozoites, merozoites, and oocytes. The *Toxoplasma* oocysts have a free-living phase as they can survive in cat feces in the environment. In contrast, DFTD cancer cells have a simple life cycle (basically cell division) with no quiescent phase and no free-living stage. Transmission of DFTD occurs directly between devils and the cancer cells will not survive in the environment. It is unlikely that DFTD can infect any other species. Marsupials that coexist with devils have never shown signs of DFTD. Furthermore, mice injected with viable DFTD cancer cells readily reject the foreign cancer cells [65].

As parasites exist within everchanging and hostile environments, natural selection shapes how parasites adapt and survive [66]. Cancer cells also live within a changing environment and need to compete for the host’s resources. Similar to parasites, cancer cells driven by genetic and epigenetic changes are continually evolving [67]. Cancer as an ongoing evolutionary process has been suggested [68]. Random genetic or epigenetic events may confer a selective advantage and drive evolutionary processes [69]. Parasitic diseases and cancer both display evolutionary traits. To detect evidence of evolution, Ujvari and colleagues investigated the methylation patterns of DFTD cells from a range of cancers that were separated by time [14]. The authors discovered complex changes in methylation patterns with DFTD. However, demethylation increased over time and this correlated with an increase in the genes for the DNA-demethylase enzymes MBD2 and MB4. The conclusion that “DFTD should not be treated as a static entity, but rather as an evolving parasite with epigenetic plasticity” [14], is one of the earliest considerations of DFTD as a parasite.

Ujvari and colleagues extended the theme that cancer and parasites have similarities. They compared the life-history traits (e.g., fecundity, survival) of parasites and cancers (including DFTD) and showed that cancers and parasites had similar effects on life history [13]. Specifically, they noted the effect of DFTD on fecundity, which increased the proportion of devils displaying precocious sexual maturity and early reproduction [70]. Cancer cells and parasites share the exploitation of the host for resources, resulting in diminishing health. It is known that a consequence of parasites exploiting host resources is an evolution of life-history traits, including reproduction and lifespan. As mentioned above, DFTD appears to have a similar effect on life-history traits. Using DFTD as a model cancer, Russell and colleagues concluded that parasites and DFTD can affect devil life-history traits [71]. The canine transmissible venereal tumor (CTVT) has been referred to as a “parasitic tumor” [72]. Another approach to comparing parasites to cancer was performed by Lun and colleagues [73]. The authors highlighted that cancer could occur in many species, including *Toxoplasma gondii*. The protozoan cancer has the potential to cause death in mammals. Parallels were identified between protozoan parasites and cancer cells. Devil facial tumor disease was used as an example of a transmissible cancer that existed as an autonomous organism and could be considered an “asexually duplicating unicellular pathogen” [73].

At the free public event in 2012, Parasite Encounters in the Wild [12], the audience was asked: “Is DFTD the perfect parasite?”. The unanimous response was “No”. While it is clear that DFTD is not a parasite, by comparing DFTD to a parasitic disease, similar features are revealed. Such a comparison may facilitate an understanding of the life cycle and mode of transmission of this unique and fascinating cancer.

## Figures and Tables

**Figure 1 tropicalmed-05-00050-f001:**
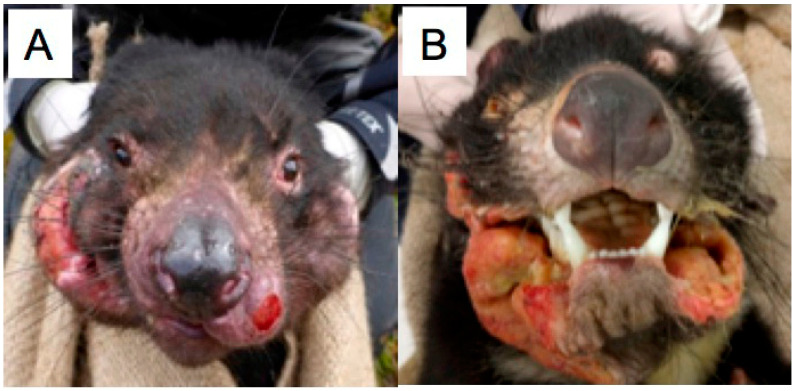
Gross facial deformities caused by (**A**) devil facial tumor disease 1 (DFT1) and (**B**) DFT2.

**Table 1 tropicalmed-05-00050-t001:** A range of immune escape strategies utilized by parasites.

Immune Escape Strategies	Parasite Example
Avoid immune recognition	*Plasmodium* spp
Quiescence	*Plasmodium* spp
Avoid phagocytosis	*Toxoplasma gondii*
Suppress the host’s immune response	*Trichinella spiralis*
Block natural killer (NK) cells	*Plasmodium falciparum*
Interfere with antigen processing	*Plasmodium* spp
Modify antigen surface identity	*Giardia Lamblia*

Adapted from [42].

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
