# Peer review of "A Devil of a Transmissible Cancer"

_tropicalmed, 2020, doi:10.3390/tropicalmed5020050_

Round 1

Reviewer 1 Report

This paper attempts to use the premise that the DFTD is actually a parasitic disease and to use this pretense as a review of the DFTD.  Throughout the paper the author creates lists of features that define parasites and then compares them to the features of the DFTD.  These characteristics are not ones that would be agreed to by any parasitologist that I know.  Additionally, the paper demonstrates a very superficial level of understanding of what a parasite is and the complexity and biology of parasites.  Bottom line, the premise of this paper is not supported by any of the “evidence” provided in it and often the “evidence” contradicts the premise.

In the end, if a review of this disease is to be accepted, then it should just be reviewed without the need to compare it to a parasite.  Also, not sure how a disease in one of the most Southern Islands in the world has anything to do with tropical diseases.

I make my point in the comments below.  I did not comment on the entire manuscript as the first part of it was sufficient to convince me that it cannot be accepted.

Line 32:  It is not necessary for a parasite to penetrate tissues to cause an infection.  Does not the presence of Giardia or round worms in the gut represent an infection, even though they do not penetrate tissue.

Line 40:  I have trouble considering cancer as a symbiont:  Symbiosis is defined as “a close, prolonged association between two or more different organisms of different species that may, but does not necessarily, benefit each member.”  First cancer is not a different organism, in most cases (with the exception of when it is caused by a virus) composed exclusively of the host genome.  Secondly, in most instances symbiosis is used to describe two organisms that mutually benefit from each others’ presence. This is definitely not the case when it comes to the DFTD.

Line 53:  There is no way that the DFTD can be considered an ideal parasite.  An ideal parasite would be one that lives in the host without causing disease or might actually be a symbiont providing some valuable service to the host, possibly by assisting in digestion of ingesta.

Lines 64 and 65:  Please combine this with the next paragraph.  It is not a standalone paragraph.

Table 1 does not make any sense to me.  Who is it, the authors, or the scientific community that have come up with this list?  Most of the things on this list that I don’t agree with as specific characteristics that define what a parasite is.  How many parasites cause disfigurement and if so, what is the pathogenesis and does it even relate to the DFTD? Leishmania caused disease as the result of inflammation, not as the result of proliferation of invading cells.  DFTD and Leishmania cannot be compared.

Line 75:  This sentence is theatrical and not appropriate for this paper.

Line 86:  This sentence needs to be referenced.  Also, it is not entirely true.  Parasites do not target weak animals.  They may be more likely to cause disease in them because of a reduced immune response or changes in behaviour (decreased grooming), but given the exposure is generally the same across healthy and unhealthy animals, this characterization of parasites is wrong.

Line 96 to 97.  The comparison between a parasite that is transmitted through biting mosquitoes where generally the parasite uses the mosquito as an intermediate host and transmission of DFTD by devils biting each other is like comparing apples to oranges.  Perhaps a better example would be rabies where disease occurs when one animal bits another.

Line 101 to 103: “There was initially no evidence for resistance, which is contrary to most parasitic diseases where some animals can survive and even recover from a parasitic infection.”  This contradicts the hypothesis of the paper and also simplifies the complex and varying relationship between hosts and their parasites.  Many parasites induce little or no host immune response and the host may not have resistance to them.  This sentence also needs to be referenced.

Line 123:  You have not convinced me that the DFTD is a parasitic disease so this line does not make any sense.  Nor does it help to prove your point.

Line 130:  Please look up the definition of contagion.  You are not using it properly in this sentence.  The fact that the DFTD is not caused by a virus does not change anything.  If it were caused by a virus and the virus had inserted an oncogene into the Devils genes, it could still behave identically to what it is doing now.  If your point is that it is not like the papillomaviruses where the virus is spread independently of the cancer that it produces, that is fine, but this point needs to be made more clearly.

Table 2 makes no sense to me.  How can you compare a transmissible tumor that avoids host recognition because it does not express MHC antigens with Mycobacteria (are they parasites or bacteria?) that live inside a cell.  Helminths are all worms.  This is a very nonspecific term and oversimplifies the complex and diverse interactions that the hundreds of thousands of helminth parasites have with their hosts.  Protozoa is also a vague term and not helpful.

Line:  163 and 164.  One needs to make a distinction between transmission, colonization, and the development of disease.  The immune status of a host may impact colonization and the development of disease but has not impact on exposure.  This is an important concept.  Also, Cryptosporidia are very capable of causing disease in healthy humans and animals and this line or argument again defeats your initial hypothesis.

Line 174:  I am sorry, but when did bacteria become parasites.  If you are going to call bacteria parasites, then you need to consider fungi and viruses as parasites so all infectious disease become parasites, so why do we call parasites parasites?  I do not find this to be logical.  

Reviewer 2 Report

This review was a delight to read. At its most fundamental it serves as a comprehensive review of the current knowledge on DFTD, but its novelty is in the argument that DFTD can be considered a parasite. Add to this a style of writing that is engaging and, as I have already said, it makes for a delightful read.

My only suggestions relate to two typos:

Line 178: missing a full-stop after reference 58

Line 228: should perhaps read as, "which increased THE PROPORTION of devils displaying precocious sexual maturity..." or something to this effect.

Author Response

Thank you for the supportive comments.

We have added the full stop after reference 58 (now 54).

We modified the Line 228 to

“Specifically, they noted the effect of DFTD on fecundity, which increased the proportion of devils displaying precocious sexual maturity and early reproduction [70].”

Reviewer 3 Report

A devil of a transmissible and parasitic cancer

The authors pretend to determinate based in definitions and some reports that the DFTV should be classified as a parasite. However it can be said that cancer cells can display a parasitic-like relationship restricted only when referring to the relationship between cancer cells and its recipient.

The manuscript is mainly based in the article of: Zhao-Rong Lun, De-Hua Lai, Yan-Zi Wen, Ling-Ling Zheng, Ji-Long Shen, Ting-Bo Yang, Wen-Liang Zhou, Liang-Hu Qu, Geoff Hide, and Francisco J. (2015).

but in contrary sense, towards DFTV. The manuscript review is not clear and it doesn't discuss about the molecular mechanisms and between the differences in cancer and parasites. In a way that seems like the proper terminology is not used correctly to define parasitism, virulence, contagious and cancer. It is required to be more conclusive in each paragraph thorough the manuscript and to change the conclusion according to the results.    

Author Response

The authors pretend to determinate based in definitions and some reports that the DFTV should be classified as a parasite. However it can be said that cancer cells can display a parasitic-like relationship restricted only when referring to the relationship between cancer cells and its recipient.

Response

It was difficult to interpret this reviewer’s comments. However, and as suggested by the editor, we have restructured the manuscript as an analogy (parasite-like relationship). We assume DFTV should be DFTD and that the "V" does not refer to "virus".

Reviewer 3

The manuscript is mainly based in the article of: Zhao-Rong Lun, De-Hua Lai, Yan-Zi Wen, Ling-Ling Zheng, Ji-Long Shen, Ting-Bo Yang, Wen-Liang Zhou, Liang-Hu Qu, Geoff Hide, and Francisco J. (2015).

Response

The Lun et al. manuscript was just one of the manuscripts quoted and was the last reference that was only quoted once in the concluding paragraphs.

Reviewer 3

but in contrary sense, towards DFTV. The manuscript review is not clear and it doesn't discuss about the molecular mechanisms and between the differences in cancer and parasites. In a way that seems like the proper terminology is not used correctly to define parasitism, virulence, contagious and cancer. It is required to be more conclusive in each paragraph thorough the manuscript and to change the conclusion according to the results.

Response

Virulence and contagious were not used in this manuscript, although three references used "contagious" in the title. Parasitism was mentioned once “Parasites are present in almost all species and parasitism is a hugely successful life form [9].”

Inclusion of the "molecular mechanisms and between the differences in cancer and parasites" as suggested, would be interesting, but would require enough space to fill the entire journal.

Reviewer 3

It is required to be more conclusive in each paragraph thorough the manuscript and to change the conclusion according to the results.

Response

We have addressed the concerns of this reviewer in our responses to reviewer  1.

Round 2

Reviewer 1 Report

This analogy just does not work and does nothing to help the understanding of Devil Facial Tumour Disease.  

Author Response

See editor's comments.

Reviewer 3 Report

 The revision manuscript is based on a simple analogy between parasitism and cancer.

The definition of parasite described in this review is not correct as it would incline any type of microorganisms.

“ however For DFTD to be considered a parasite, specific criteria must be satisfied. The Centers for Disease Control and Prevention (CDC) defined a parasite as “An organism that lives on or in a host organism and gets its food from or at the expense of its host.”

it is not clear because it is intended to create an analogy between parasitism and cancer, explain the reasons

Until date there are several theories about the generation of CTVT, that is a similar cancer in dogs transmited by sexual behavior principally (Carlo C. Maley1 and Darryl Shibata. Cancer cell evolution through the ages. AUGUST 2019 • VOL 365 ISSUE 6452. SCIENCE).

“CTVT probably started from a macrophage, which evolved into a sexually transmitted parasite that can evade the canine immune system long enough to be transmitted to a new host. It is usually cleared by the host’s immune system before it becomes lethal”

However, in order to determine this, it is necessary to determine which parasite genes are found within the DFTD and still not clarify their parasitism.

in general the analogy created between parasitism and cancer is just a terminology.

please explain this sentence in detail. “While it is clear that DFTD is not a parasite, the analogy of DFTD to a parasitic disease reveals similar features. This comparison may facilitate an understanding of the life cycle and mode of transmission of this unique and fascinating cancer.

please explain who is Dr. glodsmit and why it is important to mention it as a fundamental part of this review

“For more than 50 years Professor Goldsmid, in his memorable lectures to undergraduate students, would eloquently elaborate on the features of parasites”.

Why do this type of questioning and why is it important in your review? “At the free public event in 2012, Parasite Encounters in the Wild [12], the audience was asked: "Is DFTD the perfect parasite?". The unanimous response was "No."

I suggest that this review be rewritten and based primarily on the main framework. I suppose is what youwrite as conclusión.“DFTD to a parasitic disease reveals similar features. This comparison may facilitate an understanding of the life cycle and mode of transmission of this unique and fascinating cancer”.

Also addresing furthemore molecular aspects between parasites and DFTD, evolution, treatments and generate a perspective clear about DFTD and parasitism.

Author Response

See editor's comments.

Round 3

Reviewer 1 Report

I find the analogy that between the Devil Facial Tumour and a parasite to be unhelpful at best and absurd at its worst.  It does not help in improving the understanding of this disease or its control.  Essentially it is just a gimmick.

All one has to do is to look the images of a devil with the DFTV and the man with Leishmaniasis to know that the authors are grasping at straws to make this analogy.  Disfigurement is not a characteristic of most or even a small minority of parasitic diseases and the process in the devil is that of neoplasia where the process in this poor individual with Leishmania is inflammation and scarring.

I have now reviewed this three times and even though it has been modified to some extent and is more professionally presented, the paper still presents an argument that is unfounded and unhelpful and I cannot support its publication.  

Author Response

Leishmaniasis removed.